# Why do people resist AI-based autonomous cars?: Analyzing the impact of the risk perception paradigm and conditional value on public acceptance of autonomous vehicles

**Ie Rei Park[1], Seoyong Kim[2]\*, Jungwook Moon[3]**

**1** Department of Public Administration, Research Associate, Yonsei University, Seoul, Republic of Korea,
**2** Department of Public Administration, Ajou University, Suwon, Republic of Korea, **3** Korea Information
Society Development Institute (KISDI), Center for AI & Social Policy, Jincheon-gun, Korea

\* seoyongkim@ajou.ac.kr

## Abstract

This study examines the factors that lead to the acceptance of AI-based autonomous vehicles. Despite the considerable importance of AI-based autonomous vehicles there has been a lack of analysis based on theoretical models and analysis that considers contextual conditions. The survey was conducted between July 8 and 17, 2019. In order to increase the representativeness of the sample, a quota sampling method was adopted, based on considering gender and region. The sample size of this survey is 2,000 people. According to the response statistics, 26,231 people requested the survey, 3,973 people participated in the survey, and 2,000 people completed the survey. We adopted regression and moderation analysis as main statistical analysis methods. In modeling, we set up the variable from risk perception paradigm as independent variable and the conditional value as moderating variable in explaining the acceptance of AI-based autonomous vehicles. For this work, the analysis was conducted in two stages. Initially, a regression analysis was performed to determine the impact of the risk perception paradigm and conditional value on the acceptance of autonomous vehicles. Secondly, a moderation analysis was conducted to determine whether the perception of self-driving taxis moderates the relationship between the risk perception paradigm and the acceptance of autonomous vehicle. The study revealed that the acceptance of autonomous vehicles is influenced by a number of factors, including knowledge, image, conditional value, and perceived risks. Additionally, the relationship with perceived benefits, image and autonomous vehicle is moderated by conditional value.

## Introduction

The development of autonomous vehicles is recognized as one of the highlights of the Fourth Industrial Revolution, and progress has been rapid. Not only from a technical perspective, but also from a sociological perspective, autonomous vehicles are of great importance, especially as

**Data Availability Statement:** Data has been uploaded to figshare: 10.6084/m9.figshare. 27633435.

**Funding:** This paper is based on the research project "Advancement of Social Acceptance Model of Intelligent Information Technology and Determinants of Social Acceptance (National Research Council of Economic, Humanities, and Social Research 19-41-02)" conducted in 2019 as a collaborative research between the Information and Communication Policy Research Institute and Ajou University. This work was supported by the Ministry of Education of the Republic of Korea and the National Research Foundation of Korea (NRF-2021S1A5C2A02087244). The funders had no role in study design, data collection and analysis, decision to publish, or preparation of the manuscript.

**Competing interests:** The authors have declared that no competing interests exist.

they are closely related to human lives, such as safety and jobs. The Fourth Industrial Revolution is transforming all aspects of human life, with technology playing an increasingly significant role in tasks that were once exclusively human, such as driving. The conversation about self-driving vehicle technology is not a new one, as the idea first surfaced during the 1939 World's Fair in New York City when General Motors (GM) showcased a vehicle capable of navigating road imperfections. Robotic car is known as a self-driving car, an autonomous car, and driverless car. In the 1980s, the first self-driving car is invented by Mercedes-Benz.

As the leader in autonomous vehicles, Tesla's car sales continue to grow, as shown in Fig 1. Sales grew from just 15,510 units in 1Q16 to 479,700 units in 2Q23, a more than 30-fold increase. By 2025, By 2025, 8 million autonomous or semi-autonomous vehicles are expected to be on the road, and the global autonomous vehicle market is expected to reach 1.9 trillion by the end of 2025 [1].

One of the biggest issues with autonomous vehicles is the occurrence of accidents while driving. In 2020, there were six reported accidents involving self-driving cars in California. 16.60% of these accidents occurred when self-driving cars were being tested at low speeds. Self-driving car accidents are a major factor in reducing acceptance. However, there is an interpretation that these accidents are caused by human error rather than machine error. the Approximately 90% of significant traffic accidents resulted from human errors and negligence [2]. This could decrease as autonomous vehicles improve and reduce human intervention. There are also reports that autonomous vehicles could reduce fatal accidents by as much as 90%. Also, autonomous vehicles have a low accident rate, compared to 4.7 accidents per hundred miles for traditional vehicles. In fact, in the U.S., self-driving car testing have reduced the number of accidents per million miles from 9.1 in 2016 to 4.6 in 2018. This is projected to prevent 4.22 million accidents per year and save 21,700 lives by 2050. This reduction in accidents

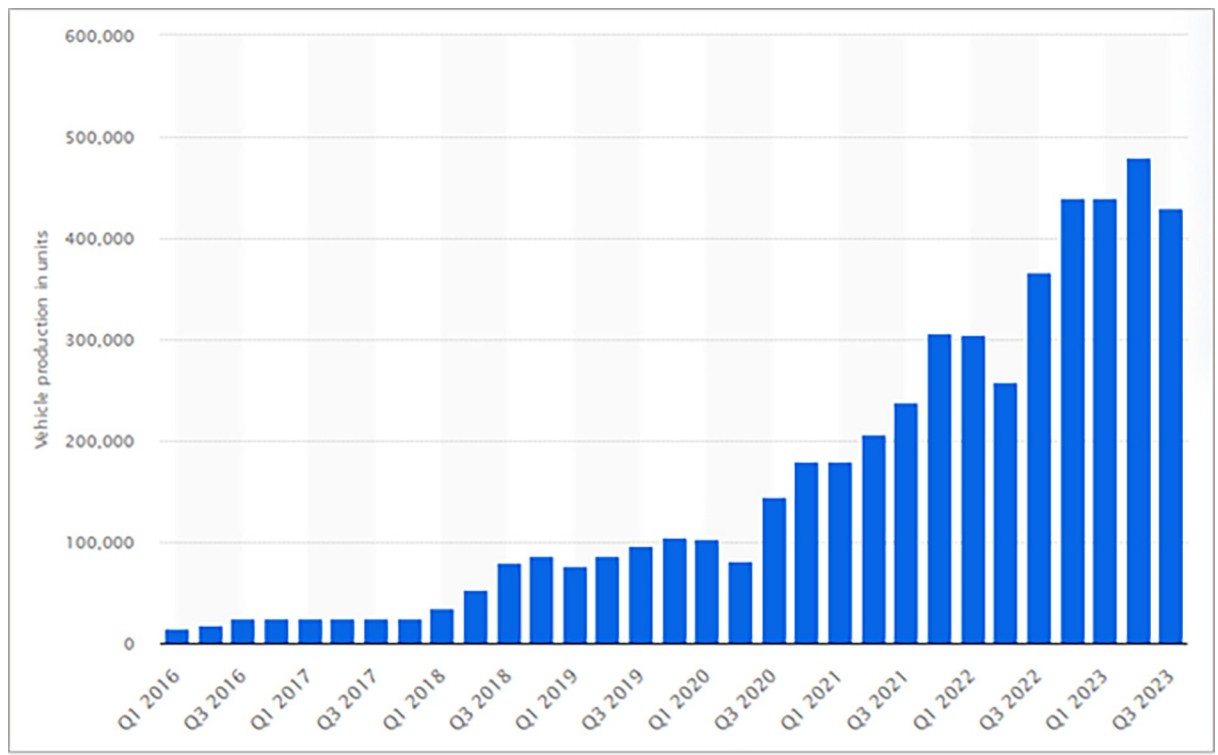

**Fig 1. Number of Tesla vehicles produced worldwide from 1st quarter 2016 to 3rd quarter 2023 (in units).**

is expected to reduce the societal cost of motor vehicle-related accidents by $65 billion, and is expected to reduce insurance costs by 60%. Autonomous vehicles could also reduce fuel consumption by 44% for trucks and 18% for cars [1].

Despite this rosy outlook, the key will be the degree to which people acceptance of autonomous vehicles. the social acceptance is considered as a main key factor for the success of any new technology. Despite the enthusiastic discussion of AVs, little is known about the public acceptance and perception of the AVs technology or the causal factors that influence the public acceptance [3]. It's worth noting that there is a causal factor in the declining acceptance of autonomous vehicles. People reveal more negative attitude toward AVs in 2017 with an average acceptance of 3.3 out of 5 than in 2014 with an average acceptance of 3.6–4.3 out of 5 in different countries [4].

Despite the drawbacks of self-driving vehicles, their advantages are so significant that their development is inevitable. The advancement of technology is important in its own right, but efforts to optimize it must also be increased to ensure appropriate development. In recent years, researchers have actively examined the acceptance of autonomous vehicles, but prior studies have predominantly concentrated on individual-level factors, including gender, age, income, and accident history. Previous research has focused primarily on individual characteristics, overlooking the theoretical model that people actually think about. In addition, there is a lack of systematic research on how conditional factors affect acceptance of autonomous vehicles. Your acceptance of self-driving cars may depend on conditional and situational factor where you are in the world. Acceptance will be very different depending on whether you are the owner of an autonomous vehicle or not, or whether you work in the autonomous vehicle industry or not. It's one thing for people to evaluate autonomous vehicles from a third-party perspective, but it's very different when it comes to their own interests. Many studies make the mistake of just asking respondents about their attitudes toward autonomous vehicles and fail to effectively connect the relationships between personal engagement and relevance. Therefore, it is necessary to study acceptance according to changing situational conditions.

The objective of this study is to identify the variables that influence the acceptance of autonomous vehicles at both the psychological and conditional levels. In order to examine the psychological dimension, we employ the risk perception paradigm, which allows us to capture an individual's subjective perception of risk. The risk perception paradigm, as proposed by Slovic et al. [1,5,6], asserts that the subjective perception of risk is dependent on the individual in question. Consequently, the acceptance of a risky object is influenced by the subjective perception of risk. The specific variables related to the risk perception paradigm include perceived risk, perceived benefit, trust, knowledge, and image. In addition, this study proposes a conditional value variable based on the theory of consumption value. The theory of consumption value is proposed by Sheth, Newman, & Gross [7] that consumers decide whether to purchase a product or not and whether to use the product or not. The conditional value of this study is used to check whether there will be a change in attitude toward the acceptance of autonomous vehicles if one assumes that one is a taxi driver. Specifically, it was conducted to check whether there is a causal relationship with the acceptance of autonomous vehicles and whether it plays a moderating role in the causal relationship between the risk perception paradigm and the acceptance of autonomous vehicles. The objective of this study is to provide policy implications for the future acceptance of autonomous vehicles and suggestions for their future development.

Such a theoretical approach can contribute to research in this area in the following ways; The model proposed in this study will contribute to the development of future AI-based acceptance models, as there have been few theoretical models what explains the acceptance of AI-based cars.

## Theoretical background and hypotheses

### Main trend in previous studies about the acceptance of autonomous cars

Recently, a lot of studies have explored variables impacting the acceptance and purchase intention of self-driving vehicles [8,9]. Previous research on autonomous vehicles has focused on the sociodemographic variable or resources owned by individuals.

Cho & Kim [10] conducted research how the socio-economic factors influence the social acceptance of autonomous vehicles. Their findings indicate that females, older individuals, and those who have been involved in automobile accidents have more positive perceptions of autonomous vehicles, and are therefore more likely to accept them. After Lee et al. [11] examined the self-driving vehicle preferences, they reported that a preference for partially and fully autonomous vehicles was higher among those who relied on private cars as their primary mode of transportation. Lavieri et al [9] showed that in terms of demographics, younger age, higher education, experience with car sharing, and residing in high-density cities were predictive of faster adoption of autonomous vehicle sharing. Haboucha et al. [8] discovered that respondents with higher education, younger age, and more time spent in the car were more accepting of self-driving vehicles. Shabanpour et al. [12] discovered individuals with higher incomes, who commute by car daily, and have a preference for new technology, are more inclined to demonstrate an interest in autonomous vehicles. Additionally, these individuals exhibit a greater willingness to pay for such technology.

In addition to demographics, a number of other variables are influencing the acceptance of autonomous vehicles. Casley et al. [13] showed that three factors, i.e., the safety of the system, the cost of the technology, and the liability issue the public acceptance of self-driving vehicles. Based on global research on driverless cars, Lavieri et al [9]conducted a survey of 1,832 individuals regarding their inclination to use self-driving cars as well as their transportation preferences and desire to own a driverless vehicle. Their research revealed that respondents who prioritize green living and embrace innovative technologies were more inclined to use autonomous vehicles. The limitations of these existing studies are that acceptance of autonomous vehicles is studied in a variable-oriented manner rather than based on a theoretical model.

Hewitt et al. [14] introduced the 'Autonomous Vehicle Acceptance Model,' which integrates the Unified Theory of Acceptance and Use of Technology (UTAUT), the Technology Acceptance Model (TAM), and the TAM2 model, drawing from various frameworks related to technology acceptance. Similarly, [15] sought to explain the intention to accept autonomous vehicles by incorporating individual-level variables and systemic characteristics into the TAM framework. In a study by Mara and Meyer [16], variables used in published research on autonomous vehicle acceptance were categorized into three main groups: user-specific determinants (e.g., demographic and personality characteristics), vehicle-specific determinants (e.g., perceived safety, predictability, and appearance), and situational determinants (e.g., road conditions).

Despite the active research on autonomous vehicle acceptance, particularly in light of the anticipated development of these technologies, existing studies face several limitations. First, much of the research is driven by specific variables rather than comprehensive theoretical models, which may lead to bias in the selection of variables. Second, although autonomous vehicles could be perceived as a threat due to social concerns such as safety issues, ethical dilemmas, and job displacement, these aspects have not been adequately explored. The introduction of new technologies often brings the potential for new risks, and Mara and Meyer [16] highlighted that psychological factors are among the key variables influencing the acceptance of autonomous vehicles. In response to these gaps, this study proposes a theoretical model

grounded in the risk perception paradigm, examining the acceptance of autonomous vehicles while considering conditional value.

## Risk perception paradigm

In particular, this study aims to examine public acceptance of autonomous vehicles according to subjective risk perception of AI-based robot technologies through the risk perception paradigm. Slovic et al [6,17,18] proposed the risk perception paradigm to gauge the subjectivity of risk perception. This paradigm is founded on four premises, the first of which asserts that risk is a subjective concept, signaling that it has diverse meanings based on the perceiver. The approach recognizes the diverse factors contributing to the assessment of risk, including technical, physical, social, and psychological aspects. The process should incorporate the insights of both experts and the public [19]. Additionally, risk judgments should be evaluated based on respondents' expressed preferences [5,20,21]. Risk perception is a subjective measure of the likelihood of a particular risk occurring, including the level of concern of the individual [22]. In traditional policy making, it is crucial to determine the level of risk objectively. Resistance toward specific risk objects arise due to conflicts around subjective risk perceptions between stakeholders, resulting in negative effects on the policy-making process [21]. In other words, when the occurrence of risk is ubiquitous, it is important to comprehend the subjectivity of risk perceptions.

We assumed that the acceptance of self-driving vehicles operate on AI-based robot technologies may vary based on recent subjective perceptions of risk of AI-based technologies. Our study categorized the general public's risk perception of the AI-based robot technologies into perceived risk, perceived benefit, trust, knowledge, and image.

## Perceived risk and perceived benefit

Perceived risk refers to an individual's evaluation of the chance of a risk happening and the potential harm and severity of the risk [23]. Perceived benefit relates to an individual's assessment of the usefulness of an object based on either tangible or intangible rewards. The interplay between perceived risk and perceived benefit is that they exhibit contrasting roles in deciding whether to accept a risky object [24]. Although there is an inverse relationship between the two variables, decreasing perceived risk does not result in an equal increase in benefits, nor do risk and benefits increase together [17,25]. The acceptance of innovative products has been negatively impacted by perceived risk, as demonstrated by previous studies [26–28]. Bauer [29] first studied perceived risk in consumer behavior research and defined it as the psychological process by which consumers perceive their own uncertainty about the outcome of a product choice. In this field, perceived risk is recognized as a subjective concept that differs from objective and probabilistic risk [30]. In general, people are highly worried about autonomous vehicle, and the level of fear of AVs increase with the rise in the number of accidents reported [3].

Perceived risk about autonomous vehicles is about the safety of the AV in poor weather and about the interaction between the vehicle and pedestrian [31] and highly concerned about the safety of the AV system [32,33]. Kim & Sung [34] determined that perceived risk negatively affected the perceived usefulness and purchase intention of autonomous vehicles. In particular, news of an accident with an autonomous vehicle can lower the positive sentiment toward autonomous vehicles [35]. Fig 2 notices that both the public fear and the number of reported accidents have the similar trend. While the number of AVs' accidents become higher, people become more feared of AVs, and while the number of AVs' accidents decline, the percentage of people fearful of AVs decrease.

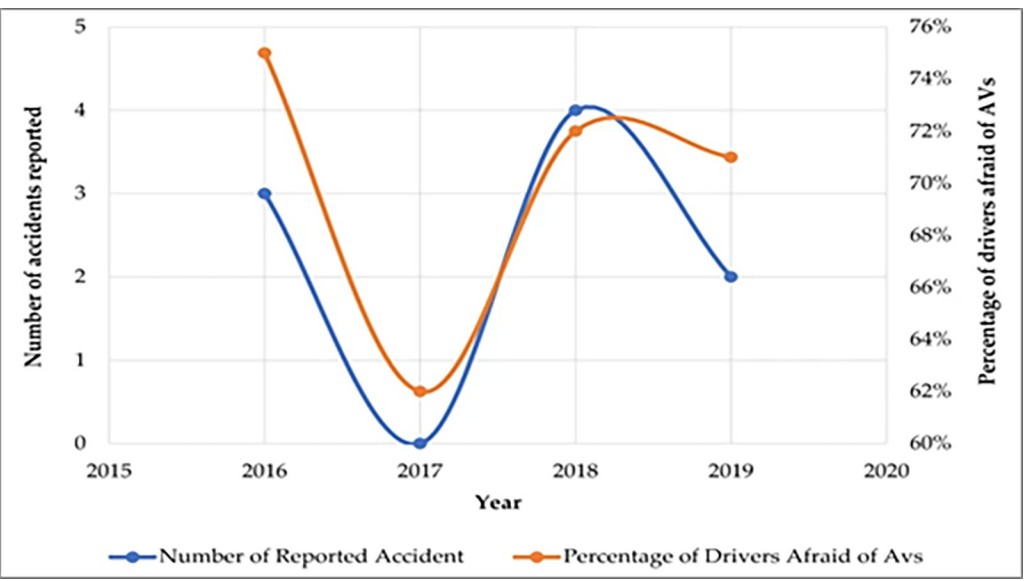

**Fig 2. Public acceptance and perception of autonomous vehicles: A comprehensive review.**

Studies on the benefits of innovative technologies have employed similar concepts such as perceived value, usefulness, and relative advantage. In the field of autonomous vehicles, perceived usefulness is a commonly utilized concept. Park et al. [36] found that perceived ease of use had a positive impact on perceived usefulness, which in turn positively influenced the intention to accept self-driving vehicles. According to their explanation, individuals perceive autonomous vehicles as useful and convenient when using them is convenient, leading to a positive impact on acceptance. Consequentialism approach stressed the benefit from autonomous vehicles. They analysed the expected utility and evaluates a set of alternatives that maximizes the overall benefit.

The studies suggest that if consumers perceive the AI-based robot technologies associated with autonomous vehicles as high-risk, it negatively affects acceptance, while a perception of high benefit has a positive effect on acceptance.

**Hypothesis 1.** A greater perceived risk of AI-based robot technologies will negatively affect the acceptance of autonomous vehicles.

**Hypothesis 2.** Greater perceived benefits of AI-based robot technologies will positively impact the acceptance of autonomous vehicles.

## Trust

Knowing the object of trust is crucial because it creates varied attitudes toward those objects [37]. Choi & Ji [38] analyzed technology acceptance by incorporating perceived risk and trust to the Technology Acceptance Model (TAM). Results exhibited a positive effect of trust on technology acceptance, and subsequent studies have measured trust in technology. These findings confirm that trust is a crucial variable in the acceptance of new technologies.

As trust decreases the presence of uncertainty in social settings, it takes a functional role in acceptance of uncertain future technology. People were asked to select the reasons that discourage them from adopting AVs. 41% of them said that they will not consider AVs because there was lack of trust in the technology [13]. Hwang & Cho [39] discovered that individuals

who had encountered speeding or traffic accidents when driving were more inclined to place trust in autonomous vehicles. Furthermore, the greater the trust, the higher the inclination to purchase self-driving vehicles. Lee et al. [11,40] also found that perceived safety played a vital role in the acceptance of self-driving vehicles, while reliability had a significant impact. This suggests that establishing trust in safety is paramount. The new about autonomous car accident decreased the level of trust in the technology especially for who have positive attitude. Finally, those people stopped tweeting about the positives of AVs [41]. Park et al. [42] observed that safety and privacy impacted trust, which in turn positively affected the intention to adopt autonomous vehicles. Park et al. [42] observed that safety and privacy impacted trust, which in turn positively affected the intention to adopt autonomous vehicles. Accordingly, increasing expectations and trust in self-driving vehicles by enhancing safety and privacy measures may boost the adoption of autonomous vehicles. Based those studies, the following hypotheses can be posited.

**Hypothesis 3.** Increased trust in AI-based robot technologies will positively influence the acceptance of autonomous vehicles.

## Knowledge

Knowledge or information reduces perceived risk of an object by resolving uncertainty and inducing acceptance [37] The general public requires time to grasp the features of a new car before purchasing it [43].

Experience about AVs will increase the knowledge about them. Based on extensive literature review, Othman [3] concluded that previous experience with AVs has a significant influence on the public acceptance of AVs. People with previous experience with AVs' features are more positive towards adopting AVs. Hutchison [44] categorized knowledge into familiarity and expertise. Familiarity denotes an individual's cognitive state of being familiar with a specific product, while expertise signifies the successful completion of tasks associated with the product. The key distinction between familiarity and expertise lies in the presence of a task: familiarity alone is insufficient for task performance, whereas expertise enables task performance through knowledge acquisition.

The first stage of the process of adopting innovations by Rogers [28] is described as knowledge. Knowledge here includes the degree of exposure to innovations, such as technical knowledge of the innovation technology, experience with the innovation technology, and information acquisition [28]. In our study, knowledge is defined as familiarity rather than the use of AI-related skills or knowledge in performing related tasks. We hypothesize that greater knowledge on AI-based robot technology and autonomous vehicles will result in a more positive acceptance towards their use due to the increased familiarity. Therefore, we posit that increased knowledge will have a positive impact on the acceptance of self-driving vehicles.

**Hypothesis 4.** Increased understanding of AI-based robot technologies could positively influence the public perception of autonomous vehicles.

## Image

Images, emotions, and feelings are utilized in cognitive psychological risk research. The correlation between images and emotions in decision-making has been identified. To measure emotions towards a specific object, the positive and negative dimensions can be evaluated as images [45] Studies have shown that positive emotions towards technology have a favorable impact on the intention to use Internet of Things (IoT) products within the context of Fourth

Industrial Revolution technologies [40]. Image plays a crucial role in technology acceptance research. In his innovation diffusion theory, Rogers [28] emphasized that during the Persuasion stage of technology acceptance, the key factor is the positive or negative perception or attitude toward the technology. Building on this idea, Davis [46–48] incorporated 'attitude' as a significant factor in his technology acceptance model. Here, attitude refers to the positive or negative perception of the technology, which can be largely influenced by the image of the technology. The image, therefore, becomes a critical determinant in shaping users' perceptions and, ultimately, their acceptance of new technologies. With respect to autonomous vehicles, Lee et al. [11] demonstrate that perceived safety, individual emotions, and trustworthiness of self-driving vehicles have a positive effect on their acceptance. It has been suggested that the policy direction should focus on enhancing safety measures while simultaneously projecting a positive image for consumers to bolster confidence in an objective manner.

**Hypothesis 5.** Positive image of AI-based robot technologies would positively influence the acceptance of autonomous vehicles.

## Theory of conditional values

Conditional value was proposed in Theory of Consumption Values, a study of the factors that lead individuals to purchase products and the values that influence their choices. Sheth et al. [7] presented Theory of Consumption Values, in which they argue that consumers' decisions to purchase or not to purchase a product, as well as to use or not to use it, are influenced by three factors: first, the reasons why consumers opt for a specific product type; second, why consumers prefer one product type over another; and third, why consumers select a certain brand over others. The three values create conditions for consumer choice, one of which is conditional value. Conditional value is the consumer's choice based on alternative outcomes in specific or varying circumstances. The empirical analysis reveals that conditional value plays a vital role in consumers' decision-making process when it comes to selecting or rejecting an item based on the presented options. According to Sheth et al. [7], conditional value is gained from any benefits that come from a specific situation where consumer faces the decision making. The TCV posits that the existence of certain external contingencies has the ability to either help or hinder consumer's performing a consumption behavior. The benefits derived from these contingencies are termed as conditional values [7]. Indeed, consumers did many decisions based on given situations, such as occasions, special life events, emergencies, festivals, and seasons [42]. Consumer research has found that changes in situational variables can significantly influence behavioral intentions [49]. In addition, a conditional value significantly influences consumers' intention to buy the pro-environmental cars [50].

To assess the acceptability of situations due to the emergence of autonomous vehicles, the variables considered the conditional value in this study. Why does the conditional value approach need? In general, experience with autonomous vehicles can have a significant impact on acceptance. 73% of respondent with previous experience with autonomous vehicles like trips by them compared with 55% for those without previous experience [51]. Thus, previous experience with autonomous vehicles has a meaningful impact on the public acceptance. However, the problem is that there are not many people with such experience in real-world. To solve this problem, it needs to consider how to assume situational conditions so that respondents can experience them indirectly. To measure the conditional value, this study conditioned on taxi drivers as included into the possible groups who are most likely to be harmed by the introduction of autonomous vehicles. The implementation and prevalent use of autonomous vehicles endanger the livelihoods of those who depend on driving for income. Wise et al [52]

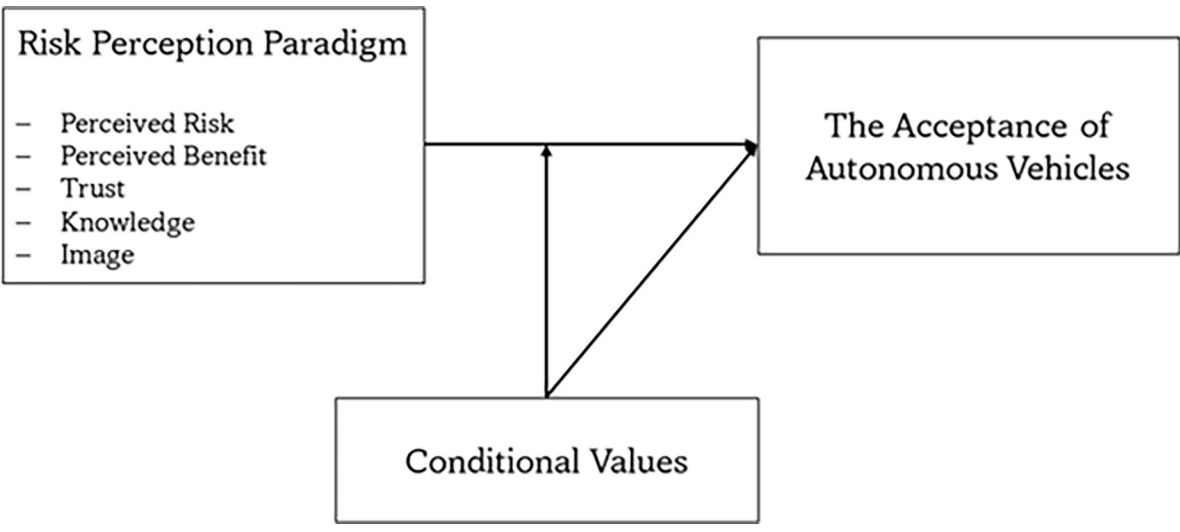

**Fig 3. Research model.**

predicted that robots, intelligent agents, or AI programs will eliminate jobs such as truck and taxi drivers, and customer service jobs in the next five years, which currently make up approximately 6% of total jobs in the US. a significant portion of truck drivers (46.2%) are concerned about the loss of driving pleasure The prediction includes the assumption that such jobs will no longer require human workers due to advancements in technology.

**Hypothesis 6.** Conditional value by putting the people into a taxi driver for autonomous vehicles will reduce the positive attitude toward of autonomous vehicles.

Additionally, this study posited that conditional value could moderate the impact of the variables in the risk perception paradigm on the acceptance of autonomous vehicles. Howard [53] acknowledged the significance of experiential learning in a given situation. Base on this, Howard & Sheth [54] developed a model of the relationship between attitude and behavior through conditional value, recognizing that neither attitude nor intention alone can accurately predict behavior. In this study also focus on acceptance of autonomous vehicles cannot be explained solely by risk-perceived attitudes. This is because individuals take into account various potential implications of the technology before accepting it.

**Hypothesis 7.** The acceptance of autonomous vehicles under conditional situation will moderate the relationship between risk perception paradigm and acceptance of autonomous vehicles.

The above discussion and hypotheses are summarized in Fig 3.

## Research design

### Data & ethics

The study collected data between July 8, 2019, and July 17, 2019, from a web survey targeting adults aged 19 and above across the general population. The sampling approach utilized proportional sampling based on regional, gender, age, and experience factors, encompassing a total of 2,000 respondents. Proportional sampling ensured that the respondents' demographics bore no significant differences. The study's participants were comprised of 48.4% males and 51.7% females. The distribution of education was 16% in their 20s, 18.2% in their 30s, 22.8% in

their 40s, 25.3% in their 50s, and 17.8% aged 60 or older. The data shows varying income and education levels among participants, with 18.1% earning less than 3 million won, 34.9% earning between 3 million won and 5 million won, and 47% earning over 5 million won. Regarding education, 34.8% of respondents held a high school diploma while 65.3% had attained a college degree or higher.

Related with ethics. We get the consent from respondent about participation in survey through written mode; respondents expressed whether or not they participates in survey and agree the information provision for research. This study did not obtain ethical approval as following reason; About ethic review committee, Bioethics and Safety Act (enacted 2005) in Korea, addressed the exemption about approval for study on human subjects. The act required the approval of the review committee in case of survey by face-to-face interviews. However, we did not face to face interview for survey; After the survey distributed the respondents, they filled out the questionnaire without anyone, and then the distributor collected the questionnaire. Moreover, article 13 of the Enforcement Decree of the Act, if research does not collect or record personally identifiable information or sensitive information in accordance with Article 23 of the 「Personal Information Protection Act」, stipulates the exemption of bio-ethic committee's review. Article 23 defined sensitive information such as personal information the opinions and beliefs, the intention to join or leave trade unions and political parties, political ideology, health, sexual life, and that may seriously infringe on other privacy of information.

## Measurement and reliability analysis

The purpose of this study is to analyze the effect of individuals' risk perception on their acceptance of autonomous vehicles. Therefore, the dependent variable is the acceptance of autonomous vehicles. The independent variables that affect acceptance are risk perception paradigm and conditional value variables, and risk perception paradigm is set as perceived risk, perceived benefit, trust, knowledge, and image. Since self-driving vehicles are based on artificial intelligence technology, the risk perception paradigm variables were set to artificial intelligence robots. Regarding the measurement items, the items used for the risk perception factors were referenced in Song & Kim [55]. we used clear, objective, and value-neutral language, avoiding biased or emotional language, ornamental language, and first-person perspectives. We utilized a passive tone and impersonal construction and employed high-level, standard language with consistent technical terms. We followed style guides and used consistent citation and footnote style and formatting features, clearly marking quotes while avoiding filler words. We maintained a formal register and avoided contractions, colloquial words, informal expressions, and unnecessary jargon. We maintained precision in word choice by using technical vocabulary where it conveyed meaning more precisely than similar non-technical terms, and we ensured grammatical correctness, spelling accuracy, and proper punctuation usage. When more than one question was measured, reliability analysis was conducted, and the mean value was used for each variable.

The study constructed the dependent variable, Autonomous vehicles as a question regarding the acceptance of autonomous vehicles. Respondents were asked to respond to two statements (How do you feel about the introduction of autonomous vehicles?;How would you fell about utilizing a self-driving vehicle for commuting or travel purposes?). The responses to these statements were measured using 5-point scale("1 = strongly disagree, 2 = slightly disagree, 3 = neutral, 4 = slightly agree, 5 = strongly agree"; Cronbach's alpha = 0.862). The independent variables were divided into Risk Perception Paradigm and Conditional Value. This study categorized variables into Perceived Risk, Perceived Benefit, Trust, Knowledge, and Image based on literature review of the Risk Perception Paradigm. Each variable was measured

using 5-point scale and the statements were as a follow: Perceived Risk(I think future domi-nated by AI robots is a disaster for humanity; I think Automated intelligent robots can be daunting; Cronbach's alpha = 0.651), Perceived Benefit(I think AI robots will contribute to solving social problems; I think AI robots will contribute to improving the quality of life; Cron-bach's alpha = 0.721), Trust(I think AI robots are very trustworthy; The companies that make AI robots are trustworthy; Cronbach's alpha = 0.831), Knowledge(I am knowledgeable about AI robots; I know more about AI robots than others; Cronbach's alpha = 0.883), Image(What are your views or thoughts on robots (Higher numbers means more positive). The study employed a conditional variable as an independent and moderating variable. Respondents were asked to provide their respond to the introduction of autonomous vehicles in the taxi industry, assuming they work as taxi drivers. Given that taxi drivers are the biggest victims of the introduction of autonomous vehicles, a positive answer to this question indicates that the respondent assesses the risk of autonomous vehicles as low.

## Results

### Descriptive statistics

The basic statistics of the variables are shown in Table 1 below. The acceptability of autono-mous vehicles is 1.28. Looking at the basic statistics of the risk perception paradigm variables, the mean of perceived risk is 2.99, perceived benefit is 3.38, trust is 3.12, knowledge is 2.65, and image is 3.18. The conditional value variable has a mean of 5.24, measured on a scale of 1 to 11.

The average of the socio-demographic variables in relation to the acceptance of Autono-mous vehicles are shown in Fig 4. To derive the average value, the responses with ten-point scale (-5 to +5) to the two response measure were converted to positive numbers (+1 to +11) and then it made mean for them. The acceptance of Autonomous vehicles of men was higher than women. The analysis of variance results showed that the difference between the two groups was statistically significant (T-value = 9.768, P-value = 0.000). This is because usually men is more interested in technology and men are likely to pursue technical careers than women. Respondents in their 20s had the highest average. The analysis of variance results showed that the difference between the five groups was statistically unsignificant (F-value = 0.391, P-value = 0.815). Although not statistically significant, the results suggest that individuals in their 20s are more interested in technology. However, the average age of car buyers in Korea is typically in their 30s, resulting in a lower average acceptance of autonomous

**Table 1. Descriptive analysis.**

| Variable | N | Minimum | Maximum | Average | SD |
|---|---|---|---|---|---|
| The Acceptance of Autonomous vehicles | 2,000 | -5 | 5 | 1.28 | 2.22 |
| Perceived Risk | 2,000 | 1 | 5 | 2.99 | 0.87 |
| Perceived Benefit | 2,000 | 1 | 5 | 3.38 | 0.70 |
| Trust | 2,000 | 1 | 5 | 3.12 | 0.72 |
| Knowledge | 2,000 | 1 | 5 | 2.65 | 0.86 |
| Image | 2,000 | 1 | 4 | 2.98 | 0.83 |
| Conditional Values | 2,000 | 1 | 11 | 5.24 | 2.70 |
| Gender(1 = Female) | 2,000 | 0 | 1 | 0.52 | 0.50 |
| Age | 2,000 | 19 | 69 | 45.43 | 13.06 |
| Income(log) | 2,000 | 0 | 8.99 | 6.03 | 0.63 |
| Education (In college or higher) | 2,000 | 0 | 1 | 0.65 | 0.48 |

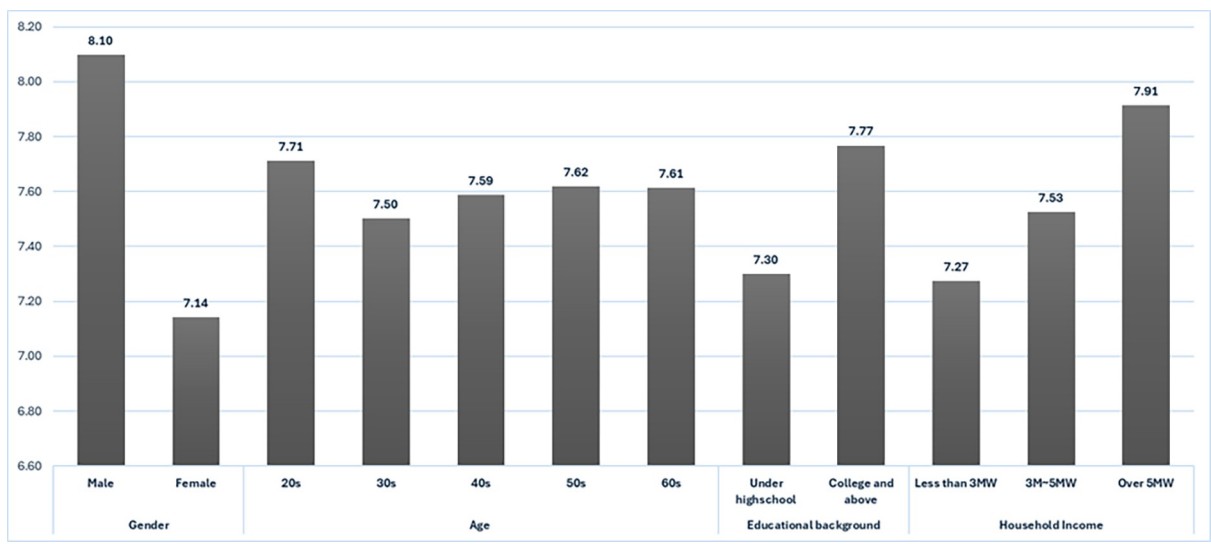

**Fig 4. Mean value of acceptance of autonomous vehicles according to socio-demographic groups.**

vehicles compared to that of individuals in their 20s. Regarding the level of education, those who were high school graduates or higher showed at acceptance of autonomous vehicles than those who were high school graduates or lower(T-value = -4.472, P-value = 0.000). This is because individuals with higher levels of education tend to possess greater knowledge about technology, which can influence their acceptance of it. This study finds that the higher the income level, higher the acceptance of autonomous vehicles (F-value = 10.765, P-value = 0.000). Autonomous vehicles are more likely to be affordable for those with higher incomes. Also, people who typically drive are in lower income brackets, which may negatively impact the adoption of autonomous vehicles.

To see the effect size, we calculated cohen's d and the results are shown in Table 2. Except for gender and education, the groups of independent variables were categorized into low group for those below the mean value and high group for those above the mean value. Education was categorized into high school or less and college or higher. The effect size is categorized as small effect size up to .3, medium effect size up to .7, and large effect size up to 1. When looking at the overall effect size, education and income have relatively small values, gender, age, risk, trust, knowledge, and conditions have medium values, and benefits, and positive image have relatively large values.

A correlation analysis was performed on autonomous vehicle acceptance, risk perception factors, and conditional value. The results are displayed in Table 3. Through this examination, it was discovered that all independent variables were significantly correlated with the dependent variable, robotic car acceptance. Perceived risk had the only negative correlation with robotic car acceptance, while perceived benefit, trust, knowledge, image, and conditional value had positive correlations.

## Regression analysis

The causal analysis results are presented in Table 4. Model 1, which excludes control variables, indicates that negative acceptance of autonomous vehicles increases with the perceived danger of AI-based robotics in the risk perception paradigm. Acceptance of self-driving vehicles was found to be positively related to perceived benefits of using AI-based robotics, increased knowledge about AI-based robotics, and a favorable image of AI-based robotics. The more

**Table 2. Anova-test and effect size.**

|  | Group | Mean | S.D. | T-value | P-value | Cohen'd | Effect-size r |
|---|---|---|---|---|---|---|---|
| Gender | Male | 1.80 | 2.22 | 105.64 | 0.00 | 0.46 | 0.23 |
|  | Female | 0.80 | 2.10 |  |  |  |  |
| Age | Low | 0.00 | 2.64 | 3.37 | 0.07 | -0.53 | -0.26 |
|  | High | 1.29 | 2.21 |  |  |  |  |
| Education | Low | 0.94 | 2.13 | 25.55 | 0.00 | -0.24 | -0.12 |
|  | High | 1.46 | 2.24 |  |  |  |  |
| Income | Low | 1.14 | 2.20 | 18.50 | 0.00 | -0.20 | -0.10 |
|  | High | 1.59 | 2.21 |  |  |  |  |
| Risk | Low | 1.76 | 2.23 | 44.96 | 0.00 | 0.32 | 0.16 |
|  | High | 1.05 | 2.17 |  |  |  |  |
| Benefit | Low | 0.49 | 2.18 | 233.74 | 0.00 | -0.69 | -0.32 |
|  | High | 1.93 | 2.02 |  |  |  |  |
| Trust | Low | 0.93 | 2.21 | 100.39 | 0.00 | -0.47 | -0.23 |
|  | High | 1.94 | 2.07 |  |  |  |  |
| Knowledge | Low | 0.92 | 2.24 | 52.01 | 0.00 | -0.32 | -0.16 |
|  | High | 1.62 | 2.14 |  |  |  |  |
| Positive image | Low | -0.52 | 2.23 | 310.06 | 0.00 | -1.07 | -0.47 |
|  | High | 1.77 | 2.03 |  |  |  |  |
| Condition | Low | 0.66 | 2.44 | 157.363 | 0.00 | -0.56 | -0.27 |
|  | High | 1.86 | 1.80 |  |  |  |  |

positive a taxi driver's attitude is toward the diffusion of autonomous vehicles, the more likely they are to accept them. Additionally, the acceptance of autonomous vehicles was found to be positively influenced by the conditional value (.293) followed by positive image (.283), perceived benefit (.208), and knowledge (.051), and negatively influenced by perceived risk (-.057). The overall explanatory power of the model is 34.6%.

The initial value of 1 for the multicollinearity test (VIF) results indicates a low correlation between variables.

Model 2 presents the outcome of a causal analysis that includes control variables. The results of the risk perception paradigm were comparable to those of Model 1. Greater perceptions of risk regarding AI-based robotics are correlated with decreased acceptance of self-driving vehicles. Conversely, more positive benefit and image of AI-based robotics are linked to

**Table 3. Correlation analysis.**

| Variables |  | 1 | 2 | 3 | 4 | 5 | 6 | 7 |
|---|---|---|---|---|---|---|---|---|
| 1.The Acceptance of Autonomous vehicles |  | 1 |  |  |  |  |  |  |
| Risk Perception Paradigm | 2. Perceived Risk | -1.55*** | 1 |  |  |  |  |  |
|  | 3. Perceived Benefit | .378*** | -.089*** | 1 |  |  |  |  |
|  | 4. Trust | .280*** | -.106*** | .482*** | 1 |  |  |  |
|  | 5. Knowledge | .221*** | -.025 | .307*** | .399*** | 1 |  |  |
|  | 6. Image | .456*** | -.228*** | .374** | .308*** | .215* | 1 |  |
| 7. Conditional Values |  | .410*** | -.073*** | .172*** | .271*** | .197*** | .107*** | 1 |

*:p < .05

**:.p < .01

***:p0 < .001.

**Table 4. Regression analysis.**

| | | B | SE | Beta | B | SE | Beta |
|---|---|---|---|---|---|---|---|
| Constant | | -5.018*** | 0.336 | | -4.565*** | 0.552 | |
| Independent | Perceived Risk | -0.143*** | 0.050 | -0.057 | -0.113** | 0.049 | -0.045 |
| | Perceived Benefit | 0.666*** | 0.073 | 0.208 | 0.641*** | 0.072 | 0.200 |
| | Trust | -0.046 | 0.072 | -0.015 | 0.088 | 0.074 | 0.029 |
| | Knowledge | 0.133** | 0.055 | 0.051 | -0.016 | 0.057 | -0.006 |
| | Positive image | 1.032*** | 0.079 | 0.283 | 0.980*** | 0.078 | 0.269 |
| | Conditional values | 0.238*** | 0.017 | 0.293 | 0.227*** | 0.016 | 0.279 |
| Control | Female | | | | -0.607** | 0.087 | -0.136 |
| | Age | | | | -0.013*** | 0.003 | -0.075 |
| | Income(log) | | | | 0.179** | 0.071 | 0.049 |
| | In college or higher | | | | 0.230** | 0.093 | 0.049 |
| F-value | | 158.122*** | | | 106.708*** | | |
| $R^2$ | | 0.346 | | | 0.373 | | |
| Adj-$R^2$ | | 0.343 | | | 0.369 | | |
| VIF | | 1.060–1.492 | | | 1.072–1.626 | | |
| Tolerance | | .670-.944 | | | .615-.938 | | |
| Durbin-Watson | | 2.069 | | | 2.092 | | |

*:$p < .05$

**:$p < .01$

***:$p < .001$.

increased favorable acceptance of autonomous vehicles. The conditional value variable indicates that taxi drivers with a favorable outlook on the proliferation of autonomous vehicles are likely to adopt them.

Additionally, all control variables significantly impact the acceptance of autonomous vehicles. Notably, being a woman or older is negatively associated with accepting autonomous vehicles, which contrasts with prior research on Korean citizens. Higher income and college education are positively associated with the acceptance of autonomous vehicles. The explanatory power of Model 2 for the ordered variables is as follows: conditional value (.279) > positive image (.269) > perceived benefit (.200) > female (-.136) > age (-.075) > > income (.049) > college education (.049) > perceived risk (-.045). Model 2 explains 37.3% of the variance, surpassing Model 1 in explanatory power.

Comparing Model 1 and Model 2, it is notable that the significant variables in the risk perception paradigm undergo a change. In Model 1, the acceptance of self-driving vehicles is not significantly influenced by trust, but knowledge has a significant positive effect. On the other hand, in Model 2, image has a significant positive effect, whereas knowledge has no significant effect. The inclusion of control variables in the analysis explains these findings.

## Moderation analysis

In this study, we used the conditional value as a moderating variable and conducted an analysis of the moderation effect. This study confirmed the moderating effect after checking for the statistical significance of the interaction term between the five variables constituting the Risk Perception Paradigm. The conditional value had a moderating effect on the relationship between the perceived benefit and the image of the robotic car acceptance variable among the five variables in the risk perception paradigm. The statistical analysis results for these two interactions

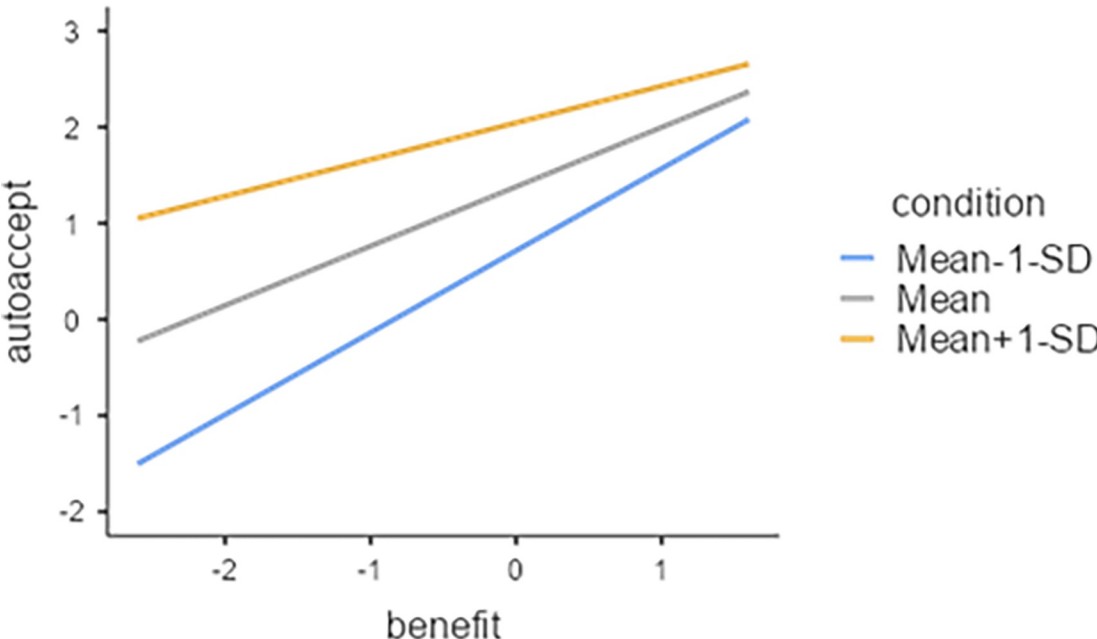

**Fig 5. Perceived benefit(IV) X Conditional value(M) = Acceptance of autonomous vehicles.**

are those in the S1 Appendix, and the contents are plotted on the simple slope graph in Figs 5–7.

Fig 5 illustrates the moderating effect of conditional value on the relationship between perceived benefit and acceptance of autonomous vehicles. As the perceived benefit increases, the acceptance of the robotic car increases, and we can see that the conditional value influences

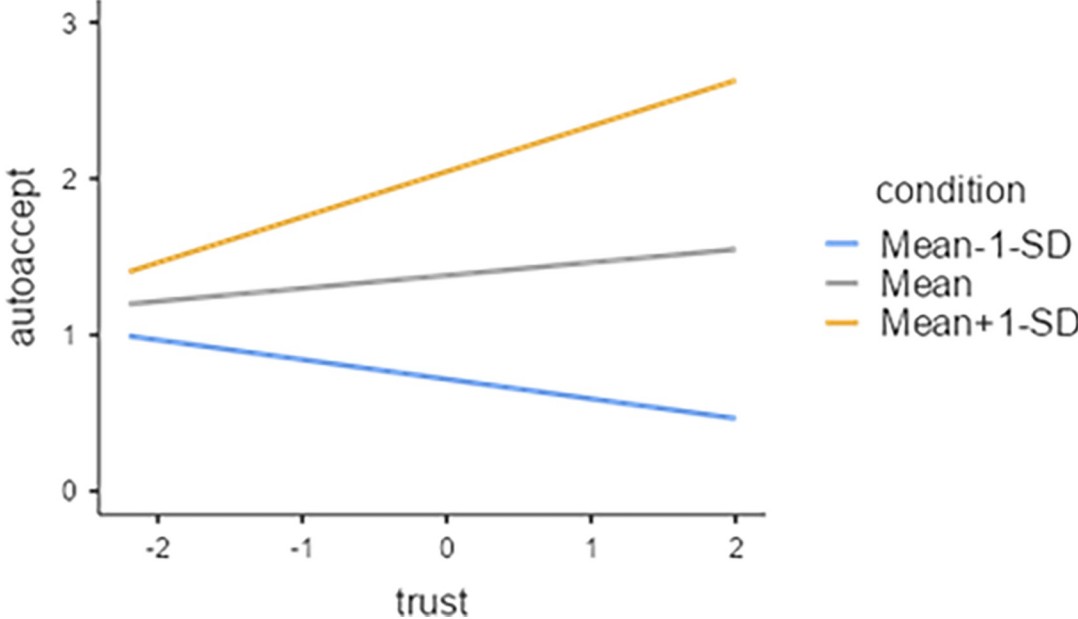

**Fig 6. Trust (IV) X Conditional value(M) = Acceptance of autonomous vehicles.**

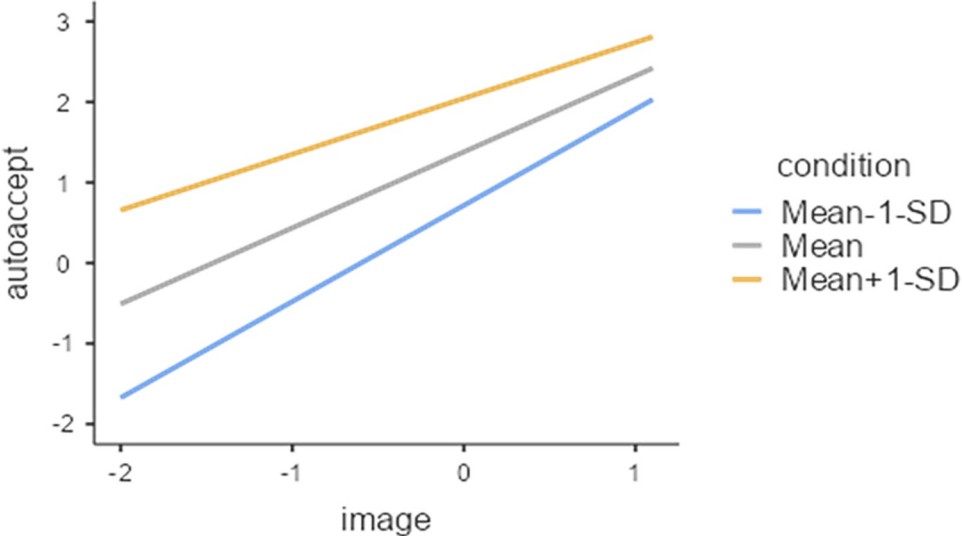

**Fig 7. Image(IV) X Conditional value(M) = Acceptance of autonomous vehicles.**

the increase. In particular, the slope of the relationship between perceived benefit those who perceived benefit higher and acceptance of autonomous vehicles is steeper in the high perceived benefit group due to the influence of the low conditional value group (those who perceive autonomous vehicles negatively when they are taxi drivers). These results show that people with high perceived benefits are more likely to accept autonomous vehicles even if they negatively affect their jobs.

Fig 6 shows that trust affects the acceptability of AI-based autonomous vehicles, and the effect is trust-dependent. Three groups can be distinguished based on the degree of trust: those who evaluate the conditional situation favorably show an increase in acceptance as trust increases. On the other hand, for those who have a negative view of the conditional situation, acceptance decreases even as trust increases. These results suggest that the positive and negative effects of trust are purely context-dependent.

Fig 7 illustrates the moderating effect of conditional value on the relationship between image of autonomous vehicles and acceptance of autonomous vehicles. We find that a favorable image of autonomous vehicles plays a positive role in the acceptance of autonomous vehicles, whereas a negative perception of autonomous vehicles plays a more positive role in the acceptance of autonomous vehicles in the conditional situation. These results suggest that negative perceptions of autonomous vehicles are offset by the positive image of robotic cards, which is very important for the acceptance of autonomous vehicles.

## Findings and discussions

This study examined the impact of risk perception and conditional value on individuals' acceptance of AI robot technologies. Perceived risk, perceived benefit, trust, knowledge, and image were used as independent variables, drawn from Slovic et al.'s [5] risk perception paradigm and Sheth et al.'s [7] consumption value theory. Technical term abbreviations will be defined upon first use. The language style adheres to academic writing conventions, ensuring objectivity and neutrality in tone. The text adheres to grammatical and spelling standards for American English. The risk perception paradigm is a valuable framework that can capture the multifaceted nature of risk. It encompasses an individual's perceived risk of an object of risk as

well as the multidimensional risk of autonomous vehicles, which can lead to various response behaviors based on differing risk perceptions. The conditional value variable proves valuable, as individual consumption choices are dependent on specific circumstances. The presence of autonomous vehicles may cause job loss, thus identifying a typical problem that can affect acceptance.

Based on the analysis, five hypotheses were formulated regarding the influence of the risk perception paradigm and conditional value on the acceptance of autonomous vehicles. Within the risk perception paradigm, perceived risk was found to negatively affect the acceptance of autonomous vehicles (Hypothesis 1). Conversely, perceived benefits and image were found to have a positive effect on the acceptance of robotic cars (Hypotheses 2, and 5). These results indicate that positive perceptions of AI technology significantly contribute to the acceptance of autonomous vehicles, suggesting the need for strategies to reduce perceived risks and negative perceptions, while simultaneously enhancing trust and positive perceptions of AI technology itself to increase public acceptance.

Hypothesis 6 was also supported, as respondents with a more optimistic view of autonomous vehicle adoption were more likely to approve of their use, as measured by a conditional value variable, which asked whether respondents would support autonomous vehicle adoption if they worked as taxi drivers. This outcome can be interpreted in light of the fact that, for taxi drivers, the introduction of autonomous vehicles may reduce their technical role, thereby lowering their workload. Furthermore, the conditional value variable partially supported Hypothesis 7 by moderating the relationship between perceived benefits, image, and acceptance of robotic cars within the risk perception paradigm.

The analytical outcomes varied depending on the inclusion of demographic variables as control factors. In Model 1, where control variables were excluded, greater knowledge of AI technologies positively influenced the acceptance of autonomous vehicles, while trust had no significant effect. However, in Model 2, which incorporated demographic control variables, the positive influence of knowledge on acceptance became insignificant, while trust became both positive and significant. Given that autonomous vehicles are part of the Fourth Industrial Revolution, knowledge of AI-related technologies is likely to be higher among individuals with higher levels of education. Therefore, it is plausible that the significance of the knowledge variable was attenuated by the inclusion of control variables such as income and education, which are likely correlated with knowledge levels. Similarly, the insignificance of the trust variable in Model 1, which became significant in Model 2, can be explained by the general increase in trust that accompanies specialized knowledge in a particular field. In Model 1, the effect of knowledge may have overshadowed trust, but in Model 2, the influence of knowledge was controlled, allowing trust to emerge as a significant factor in the acceptance of autonomous vehicles.

This study contributes to the existing body of literature by examining the impact of psychological factors on the acceptance of autonomous vehicles. Whereas prior research has largely focused on demographic, economic, and technological dimensions, this study incorporates control variables from previous studies to assess the role of psychological factors in shaping public acceptance.

## Conclusion and limits

This study employed regression analysis and moderation analysis, incorporating conditional value variables related to the risk perception paradigm and situational conditions that may influence the adoption of autonomous vehicles, with the aim of identifying psychological factors that affect their acceptance. The analysis revealed that within the risk perception

paradigm, perceived risk exerted a negative influence on the acceptance of autonomous vehicles, whereas perceived benefits, trust, and image had a positive influence. Furthermore, conditional value grounded in bounded rationality demonstrated a positive effect on the acceptance of autonomous vehicles and moderated the relationship between perceived benefits, image, and their acceptance.

This study differs from previous research by attempting to explain the acceptance of autonomous vehicles from a psychological perspective, particularly focusing on individual risk perception, rather than solely relying on the technology acceptance model, which has been the primary focus of earlier studies.

Based on our findings, it is necessary to reduce the perceived risk associated with AI-based robot technologies, which was found to have a negative impact on the acceptance of autonomous vehicles, while increasing trust, which was found to have a positive impact on acceptance. This can be achieved by ensuring the safety of autonomous vehicles through the establishment of safety standards, policy guidelines, and technical safety measures (Kim, 2018). Efforts should be made to establish responsibility in case of accidents caused by autonomous vehicles.

Second, perceived benefits, image, and trust were found to have a strong positive effect on the acceptance of autonomous vehicles. This finding underscores the importance of fostering positive psychological dimensions in enhancing the acceptance of autonomous vehicles. Therefore, promoting the benefits and positive image associated with autonomous vehicles is essential. However, the crux of fostering a positive perception lies in whether autonomous vehicles can adequately secure reliability at a technical level. To enhance reliability, it is imperative to address and eliminate risk factors, both direct and indirect, such as improving the safety of autonomous vehicles, reducing the likelihood of accidents, and resolving ethical dilemmas.

Third, the study observed that conditional value had a significant positive effect on the acceptance of autonomous vehicles. This suggests that a favorable attitude toward autonomous vehicles, particularly when considered from the perspective of taxi drivers, can strengthen their acceptance. This result can be interpreted in two ways. First, it may indicate that at the time of the study, there was a high level of trust in autonomous vehicles, which positively influenced their acceptance. Second, it suggests that the expected reduction in taxi drivers' workload due to autonomous vehicles contributed to this positive effect. It is widely acknowledged that the rise of AI technology poses a threat to the continuity and stability of workers' jobs [56]. Taxi drivers, in particular, are often highlighted as a profession at high risk of job replacement by autonomous vehicles. However, it is also anticipated that the advancement of AI technology will not necessarily result in job displacement but could instead lead to a coexistence of AI technology and human workers [57,58]. In reality, the complete replacement of jobs by AI technology remains politically, socially, and economically uncertain [59]. Moreover, AI technology has never operated entirely independently without human oversight [60], and even if it were to do so, it would still face unavoidable ethical dilemmas. These ethical dilemmas extend beyond job replacement to encompass all potential ethical issues that may arise from the deployment of AI technology. This study confirms that there is an expectation among people for the coexistence of workers and AI technology, rather than the outright replacement of jobs through AI advancements. This finding suggests a potential direction for the development of AI technology, including autonomous vehicles, where human-AI collaboration is emphasized.

Fourth, the conditional value variable had a moderating effect on the relationship between perceived benefits and image and acceptance of AI-based autonomous vehicles. We set up conditional question 'if you were a taxi driver, would you favor the introduction of self-driving cars?' A higher value indicates a higher level of favor toward AI-based autonomous vehicles.

Taxi drivers are likely to be the most affected by the widespread adoption of autonomous vehicles. Therefore, conditional effect implies that a positive response to this question suggests that the respondent perceives the risks associated with autonomous vehicles to be minimal.

In this study, we found that the effect of trust is context-dependent, i.e., for acceptance to increase when trust increases, positive evaluations of the condition must be dominant. Conversely, when the condition is dominated by negative effects, acceptance decreases even when trust increases. These results suggest that in order to increase acceptance of AI-based autonomous vehicles, it is important to use strategies to increase trust, but also to consider the Rocontextual aspects of acceptance.

Increased perceived benefits and positive emotions are associated with greater acceptance of AI-based autonomous vehicles, but this effect is conditional. The positive impact of perceived benefits and positive emotions is the case when there is a higher condition effect. This means that even if the benefits and image of autonomous vehicles are good, acceptance of AI-based autonomous vehicles is higher when there is less damage to the situation involved.

In particular, the group with high perceived benefits and image showed a high increase in the acceptance despite the low value of conditional value. This can be explained as the 'persuasion' stage, which is the stage before the acceptance of innovation, among Rogers' five stages of innovation diffusion. Persuasion involves being receptive to an innovation and gaining awareness before deciding to accept it. If the benefits of adopting the technology or the perception of the technology are positive, the individual progresses to the stage of accepting the technology [28]. If an individual perceives the benefits of adopting a technology positively, they are more likely to progress to the technology acceptance stage [28].

The implications of these findings are that, first, in order to increase the acceptance of AI-based autonomous vehicles, it is necessary to pursue ways to minimize industrial damage at the structural level, not only to increase utility. Second, the acceptance issue is not just an individual choice problem, but also a social group choice problem, which means that it is necessary to consider the class of workers who are affected by the development of science and technology. Third, the influence of conditional variables is greater in situations where perceived benefits and positive emotions are low. This suggests that in the negative context of AI-based autonomous vehicles, more consideration should be given to social side effects and harms, especially in the context of negative perceptions of AI-based autonomous vehicles.

The use of conditional values assumes certain situations that respondents may not experience. Therefore, it can be inferred that respondents' positive perception or expected benefits of AI-based robot cars had a positive effect on acceptance, even if they held negative views about the car in a conditional situation.

Although autonomous vehicles have the potential to simplify transportation for individuals, the implementation of these vehicles on a large scale will require significant changes to current road standards. Furthermore, the development of autonomous vehicles will inevitably lead to job losses for those who earn their livelihood from driving. Therefore, policymakers must carefully consider the societal impact of these technologies in addition to their potential benefits. While the study results suggest that inducing positive attitudes toward acceptance is possible, it has limitations in considering other relevant issues. Furthermore, the average response to whether respondents, as taxi drivers, would agree to autonomous vehicle deployment indicates a need for policy-level considerations regarding job displacement. The development stage of autonomous vehicles may significantly influence the selection of these conditional values.

This study examined variables influencing the acceptance of autonomous vehicles, anticipated through advancements in AI technology. However, several limitations should be noted. First, the study is susceptible to common source bias, which arises from using a single survey dataset to measure variables. This approach limits the ability to capture changes over time or

in response to policy shifts, as cross-sectional data collected at a single point in time was employed. To address this limitation, future research should consider longitudinal studies that can track changes in AI technology risk perception and the evolving acceptance of autonomous vehicles over time.

Second, the conditional value variables in this study require clearer design. If the potential for job loss were explicitly integrated into the items used for conditional variables, respondents' decision-making based on bounded rationality would likely be more evident. Future research should clearly present and incorporate factors that may be perceived as individual threats, such as job loss, from a social and ethical perspective.

Third, this study demonstrated that subjective risk perception influences the acceptance of autonomous vehicles. However, future studies should also aim to measure the perception of actual risks that are expected to materialize. By identifying which risks do or do not impact acceptance, future research can contribute to improving acceptance through comprehensive risk management strategies.

## Supporting information

**S1 Appendix.**
(DOCX)

## Author Contributions

**Conceptualization:** Seoyong Kim.

**Data curation:** Seoyong Kim.

**Investigation:** Ie Rei Park.

**Methodology:** Ie Rei Park.

**Resources:** Ie Rei Park.

**Writing – original draft:** Jungwook Moon.

**Writing – review & editing:** Seoyong Kim.

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
