## [Decision Letter · Decision Letter 0]

14 Aug 2024

PONE-D-24-30082Why Do People Resist AI-based Robotic Cars?: Analyzing the Impact of the Risk Perception Paradigm and Conditional Value on Public Acceptance of Autonomous Vehicles.PLOS ONE

Dear Dr. Kim,

Thank you for submitting your manuscript to PLOS ONE. After careful consideration, we feel that it has merit but does not fully meet PLOS ONE’s publication criteria as it currently stands. Therefore, we invite you to submit a revised version of the manuscript that addresses the points raised during the review process.

We look forward to receiving your revised manuscript.

Kind regards,

Zeashan Hameed Khan, Ph.D.

Academic Editor

PLOS ONE

Journal Requirements:

"This paper is based on the research project "Advancement of Social Acceptance Model of Intelligent Information Technology and Determinants of Social Acceptance (National Research Council of Economic, Humanities, and Social Research 19-41-02)" conducted in 2019 as a collaborative research between the Information and Communication Policy Research Institute and Ajou University. This work was supported by the Ministry of Education of the Republic of Korea and the National Research Foundation of Korea (NRF-2021S1A5C2A02087244)."

4. Please note that funding information should not appear in the Acknowledgments section or other areas of your manuscript. We will only publish funding information present in the Funding Statement section of the online submission form. Please remove any funding-related text from the manuscript. 

5. Please provide a complete Data Availability Statement in the submission form, ensuring you include all necessary access information or a reason for why you are unable to make your data freely accessible. If your research concerns only data provided within your submission, please write "All data are in the manuscript and/or supporting information files" as your Data Availability Statement.

7. PLOS requires an ORCID iD for the corresponding author in Editorial Manager on papers submitted after December 6th, 2016. Please ensure that you have an ORCID iD and that it is validated in Editorial Manager. To do this, go to ‘Update my Information’ (in the upper left-hand corner of the main menu), and click on the Fetch/Validate link next to the ORCID field. This will take you to the ORCID site and allow you to create a new iD or authenticate a pre-existing iD in Editorial Manager. Please see the following video for instructions on linking an ORCID iD to your Editorial Manager account: https://www.youtube.com/watch?v=_xcclfuvtxQ

8. Please include your full ethics statement in the ‘Methods’ section of your manuscript file. In your statement, please include the full name of the IRB or ethics committee who approved or waived your study, as well as whether or not you obtained informed written or verbal consent. If consent was waived for your study, please include this information in your statement as well. 

9. Please ensure that you refer to Figure 3 in your text as, if accepted, production will need this reference to link the reader to the figure.

Reviewers' comments:

Reviewer's Responses to Questions

**Comments to the Author**

1. Is the manuscript technically sound, and do the data support the conclusions?

Reviewer #1: No

Reviewer #2: Yes

2. Has the statistical analysis been performed appropriately and rigorously? 

Reviewer #1: No

Reviewer #2: Yes

3. Have the authors made all data underlying the findings in their manuscript fully available?

Reviewer #1: Yes

Reviewer #2: No

4. Is the manuscript presented in an intelligible fashion and written in standard English?

Reviewer #1: Yes

Reviewer #2: Yes

5. Review Comments to the Author

Reviewer #1: 1. The abstract mentions the use of the "risk perception paradigm and conditional value" but does not clarify how these theoretical models were specifically applied in the analysis. Providing more detail on how these frameworks were operationalized would strengthen the abstract's clarity and rigor.

2. The abstract lacks information on the methodology used to explore the causal relationships, such as the sample size, data collection methods, or statistical techniques employed. Including these details would provide better insight into the robustness and validity of the findings

3. The introduction should clearly conclude with a distinct section highlighting the novel contributions of your work.

4. The literature review could benefit from more explorations of previous studies. This will provide a richer context and demonstrate how your work builds upon or diverges from established research.

5. The discussion section needs to be expanded to more thoroughly analyze the results.

6. Zero visualization efforts.

7. I can not help but notice the name of such figures, as 3, are not academic. Please revise the entire manuscript for such similarity.

8. The first paragraph of the conclusion should succinctly summarize the contributions of the study in past tense, clearly stating what has been accomplished and the impact it has on the field.

9. The second paragraph of the conclusion should provide clear and actionable future recommendations. This will guide subsequent research and highlight potential areas for further investigation based on your findings

Reviewer #2: Summary: The article provides a timely analysis of the factors that influence public acceptance of autonomous vehicles, focusing on risk perception and conditional value. The study is grounded in a solid theoretical framework and offers valuable insights into the psychological and contextual factors that shape public attitudes toward the self-driving car technology.

Strengths:

* The use of the risk perception paradigm and conditional value theory is a strong aspect.

* The literature review is thorough and effectively identifies the gaps in existing research.

* The study’s empirical approach, with a large sample size of 2,000 respondents is extensive.

* The practical implications for policymakers and industry stakeholders are also well-articulated.

Weaknesses:

* The development of the hypotheses could be more clearly linked to the theoretical framework, particularly in explaining how knowledge influences risk perception.

* The moderation analysis can benefit from a more detailed discussion of the results and their implications, especially regarding different demographic groups.

* The study’s reliance on cross-sectional data limits the ability to draw causal inferences. While this limitation is acknowledged, the discussion would be enhanced by suggesting how future research could address this issue, perhaps through longitudinal studies.

* The discussion on the ethical implications of autonomous vehicles could be expanded to include considerations beyond job displacement, such as data privacy and social inequality.

* In statistical analysis, providing effect sizes alongside p-values would also help readers understand the practical significance of the findings.

6. PLOS authors have the option to publish the peer review history of their article (what does this mean?). If published, this will include your full peer review and any attached files.

Reviewer #1: **Yes: **Luttfi A. Al-Haddad

Reviewer #2: No

---

## [Author Response · Author response to Decision Letter 0]

27 Sep 2024

Pease, see the attached revision note.

Comment from Reviewer 1 and Revision

Thank you for the kind reviews. What comments from reviewer was very helpful for the development of the paper. We tried to revise all of things that the reviewer pointed out. Once again, if reviewer gives us a good comment, we will use it to improve our paper. 

Best regards, 

Authors

□ Reviewer #1

□ Comment 1: The abstract mentions the use of the "risk perception paradigm and conditional value" but does not clarify how these theoretical models were specifically applied in the analysis. Providing more detail on how these frameworks were operationalized would strengthen the abstract's clarity and rigor.

○ Answer: The reason for choosing 11 values was discussed in detail as follows and added to the text. 

○ Revision: We explained how we analyzed the theoretical model of this study in the abstract.

We set up the variable from risk perception paradigm as independent variable and the conditional value as moderating variable in explaining the acceptance of AI-based autonomous vehicles. For this work, the analysis was conducted in two stages. Initially, a regression analysis was performed to determine the impact of the risk perception paradigm and conditional value on the acceptance of autonomous vehicles. Secondly, a moderation analysis was conducted to determine whether the perception of self-driving taxis moderates the relationship between the risk perception paradigm and the acceptance of autonomous vehicle.

The study revealed that the acceptance of autonomous vehicles is influenced by a number of factors, including knowledge, image, conditional value, and perceived risks. Additionally, the relationship with perceived benefits, image and autonomous vehicle is moderated by conditional value.

□ Comment 2: The abstract lacks information on the methodology used to explore the causal relationships, such as the sample size, data collection methods, or statistical techniques employed. Including these details would provide better insight into the robustness and validity of the findings.

○ Answer: We have briefly added the research design section to the abstract. 

○ Revision: 

The survey was conducted between July 8 and 17, 2019. In order to increase the representativeness of the sample, a quota sampling method was adopted, based on considering gender and region. The sample size of this survey is 2,000 people. The maximum allowable sampling error is ±2.2 percentage points at the 95% confidence level, assuming random sampling. The survey method was a web survey. The URL was sent via mobile phone text and email, and then respondents answered a self-completion survey. According to the response statistics, 26,231 people requested the survey, 3,973 people participated in the survey, and 2,000 people completed the survey. The participation rate is 7.6% of requests. We adopted regression and moderation analysis as main statistical analysis methods. 

□ Comment 3: The introduction should clearly conclude with a distinct section highlighting the novel contributions of your work.

○ Answer: We briefly add theory to the introduction to highlight what can be gained from this study.

○ Revision: 

The objective of this study is to identify the variables that influence the acceptance of autonomous vehicles at both the psychological and conditional levels. In order to examine the psychological dimension, we employ the risk perception paradigm, which allows us to capture an individual's subjective perception of risk. The risk perception paradigm, as proposed by Slovic et al. (1981), asserts that the subjective perception of risk is dependent on the individual in question. Consequently, the acceptance of a risky object is influenced by the subjective perception of risk. The specific variables related to the risk perception paradigm include perceived risk, perceived benefit, trust, knowledge, and image. In addition, this study proposes a conditional value variable based on the theory of consumption value. The theory of consumption value is proposed by Sheth, Newman, & Gross (1991) that consumers decide whether to purchase a product or not and whether to use the product or not. The conditional value of this study is used to check whether there will be a change in attitude toward the acceptance of autonomous vehicles if one assumes that one is a taxi driver. Specifically, it was conducted to check whether there is a causal relationship with the acceptance of autonomous vehicles and whether it plays a moderating role in the causal relationship between the risk perception paradigm and the acceptance of autonomous vehicles. The objective of this study is to provide policy implications for the future acceptance of autonomous vehicles and suggestions for their future development. 

Such a theoretical approach can contribute to research in this area in the following ways; The model proposed in this study will contribute to the development of future AI-based acceptance models, as there have been few theoretical models what explains the acceptance of AI-based cars.

□ Comment 4: The literature review could benefit from more explorations of previous studies. This will provide a richer context and demonstrate how your work builds upon or diverges from established research.

○ Answer: We added literature review supplemented: Summary of autonomous vehicle research, limitations of previous studies, and theoretical contribution of knowledge and trust. 

○ Revision: 

(Page 5)Hewitt et al. (2019) introduced the 'Autonomous Vehicle Acceptance Model,' which integrates the Unified Theory of Acceptance and Use of Technology (UTAUT), the Technology Acceptance Model (TAM), and the TAM2 model, drawing from various frameworks related to technology acceptance. Similarly, Nastjuk (2020) sought to explain the intention to accept autonomous vehicles by incorporating individual-level variables and systemic characteristics into the TAM framework. In a study by Mara and Meyer (2022), variables used in published research on autonomous vehicle acceptance were categorized into three main groups: user-specific determinants (e.g., demographic and personality characteristics), vehicle-specific determinants (e.g., perceived safety, predictability, and appearance), and situational determinants (e.g., road conditions). 

Despite the active research on autonomous vehicle acceptance, particularly in light of the anticipated development of these technologies, existing studies face several limitations. First, much of the research is driven by specific variables rather than comprehensive theoretical models, which may lead to bias in the selection of variables. Second, although autonomous vehicles could be perceived as a threat due to social concerns such as safety issues, ethical dilemmas, and job displacement, these aspects have not been adequately explored. The introduction of new technologies often brings the potential for new risks, and Mara and Meyer (2022) highlighted that psychological factors are among the key variables influencing the acceptance of autonomous vehicles. In response to these gaps, this study proposes a theoretical model grounded in the risk perception paradigm, examining the acceptance of autonomous vehicles while considering conditional value.

(Page 9)The first stage of the process of adopting innovations by Rogers (1983) is described as knowledge. Knowledge here includes the degree of exposure to innovations, such as technical knowledge of the innovation technology, experience with the innovation technology, and information acquisition (Rogers, 1983).

(Page 9)Image plays a crucial role in technology acceptance research. In his innovation diffusion theory, Rogers (1983) emphasized that during the Persuasion stage of technology acceptance, the key factor is the positive or negative perception or attitude toward the technology. Building on this idea, Davis (1985) incorporated ‘attitude’ as a significant factor in his technology acceptance model. Here, attitude refers to the positive or negative perception of the technology, which can be largely influenced by the image of the technology. The image, therefore, becomes a critical determinant in shaping users' perceptions and, ultimately, their acceptance of new technologies.

+ Hewitt, C., Theocharis, A., Advait, S & Ioannis, P. Assessing Public Perception of Self-Driving Cars: the Autonomous Vehicle Acceptance Model. 2019. 10.1145/3301275.3302268.

+ Mara, M., Meyer, K. Acceptance of Autonomous Vehicles: An Overview of User-Specific, Car-Specific and Contextual Determinants. In: Riener, A., Jeon, M., Alvarez, I. (eds) User Experience Design in the Era of Automated Driving. Studies in Computational Intelligence, vol 980. Springer, Cham: 2022. https://doi.org/10.1007/978-3-030-77726-5_3

+ Nastjuk, I., Herrenkind, B., Marrone, M., Brendel, A, B., Kolbe, L, M. What drives the acceptance of autonomous driving? An investigation of acceptance factors from an end-user's perspective, Technological Forecasting and Social Change, 2020; Volume 161. https://doi.org/10.1016/j.techfore.2020.120319.

+ Othman, K. Public acceptance and perception of autonomous vehicles: a comprehensive review. AI Ethics. 2021; 1, 355–387. https://doi.org/10.1007/s43681-021-00041-8. 

+ Rogers. Diffusion of Innovations. 1983. New York: Free Press. (Original work published 1962).

□ Comment 5: The discussion section needs to be expanded to more thoroughly analyze the results.

○ Answer: We have added some of the discussion-worthy content from the analysis results.

○ Revision: 

Based on the analysis, five hypotheses were formulated regarding the influence of the risk perception paradigm and contingent value on the acceptance of autonomous vehicles. Within the risk perception paradigm, perceived risk was found to negatively affect the acceptance of autonomous vehicles (Hypothesis 1). Conversely, perceived benefits, trust, and image were found to have a positive effect on the acceptance of robotic cars (Hypotheses 2, 3, and 5). These results indicate that positive perceptions of AI technology significantly contribute to the acceptance of autonomous vehicles, suggesting the need for strategies to reduce perceived risks and negative perceptions, while simultaneously enhancing trust and positive perceptions of AI technology itself to increase public acceptance.

Hypothesis 6 was also supported, as respondents with a more optimistic view of autonomous vehicle adoption were more likely to approve of their use, as measured by a conditional value variable, which asked whether respondents would support autonomous vehicle adoption if they worked as taxi drivers. This outcome can be interpreted in light of the fact that, for taxi drivers, the introduction of autonomous vehicles may reduce their technical role, thereby lowering their workload. Furthermore, the conditional value variable partially supported Hypothesis 7 by moderating the relationship between perceived benefits, image, and acceptance of robotic cars within the risk perception paradigm.

The analytical outcomes varied depending on the inclusion of demographic variables as control factors. In Model 1, where control variables were excluded, greater knowledge of AI technologies positively influenced the acceptance of autonomous vehicles, while trust had no significant effect. However, in Model 2, which incorporated demographic control variables, the positive influence of knowledge on acceptance became insignificant, while trust became both positive and significant. Given that autonomous vehicles are part of the Fourth Industrial Revolution, knowledge of AI-related technologies is likely to be higher among individuals with higher levels of education. Therefore, it is plausible that the significance of the knowledge variable was attenuated by the inclusion of control variables such as income and education, which are likely correlated with knowledge levels. Similarly, the insignificance of the trust variable in Model 1, which became significant in Model 2, can be explained by the general increase in trust that accompanies specialized knowledge in a particular field. In Model 1, the effect of knowledge may have overshadowed trust, but in Model 2, the influence of knowledge was controlled, allowing trust to emerge as a significant factor in the acceptance of autonomous vehicles.

This study contributes to the existing body of literature by examining the impact of psychological factors on the acceptance of autonomous vehicles. Whereas prior research has largely focused on demographic, economic, and technological dimensions, this study incorporates control variables from previous studies to assess the role of psychological factors in shaping public acceptance.

□ Comment 6: Zero visualization efforts.

○ Answer: We changed the figure and added to contribution of our findings in conclusion. Please, give us tip for Zero visualization methods. 

□ Comment 7: I can not help but notice the name of such figures, as 3, are not academic. Please revise the entire manuscript for such similarity.

○ Answer: Based on extensive theoretical review, Robotic cars were changed to Autonomous Vehicles, and variable names were modified overall.

○ Revision: 

□ Comment 8: The first paragraph of the conclusion should succinctly summarize the contributions of the study in past tense, clearly stating what has been accomplished and the impact it has on the field.

○ Answer: We did the first paragraph of the conclusion summarizes the research results and describes the differences from previous studies. 

○ Revision: 

This study employed regression analysis and moderation analysis, incorporating conditional value variables related to the risk perception paradigm and situational conditions that may influence the adoption of autonomous vehicles, with the aim of identifying psychological factors that affect their acceptance. The analysis revealed that within the risk perception paradigm, perceived risk exerted a negative influence on the acceptance of autonomous vehicles, whereas perceived benefits, trust, and image had a positive influence. Furthermore, conditional value grounded in bounded rationality demonstrated a positive effect on the acceptance of autonomous vehicles and moderated the relationship between perceived benefits, image, and their acceptance.

This study differs from previous research by attempting to explain the acceptance of autonomous vehicles from a psychological perspective, particularly focusing on individual risk perception, rather than solely relying on the technology acceptance model, which has been the primary focus of earlier studies.

□ Comment 9: The second paragraph of the conclusion should provide clear and actionable future recommendations. This will guide subsequent research and highlight potential areas for further investigation based on your findings.

○ Answer: We have added the following future recommendations

○ Revision: 

Second, perceived benefits, image, and trust were found to have a strong positive effect on the acceptance of autonomous vehicles. This finding underscores the importance of fostering positive psychological dimensions in enhancing the acceptance of autonomous vehicles. Therefore, promoting the benefits and positive image associated with autonomous vehicles is essential. However, the crux of fostering a positive perception lies in whether autonomous vehicles can adequately secure reliability at a technical level. To enhance reliability, it is imperative to address and eliminate risk factors, both direct and indirect, such as improving the safety of autonomous vehicles, reducing the likelihood of accidents, and resolving ethical dilemmas.

Third, the study observed that conditional value had a significant positive effect on the acceptance of autonomous vehicles. This suggests that a favorable attitude toward autonomous vehicles, particularly when considered from the perspective of taxi drivers, can strengthen their acceptance. This result can be interpreted in two ways. First, it may indicate that at the time of the study, there was a high level of trust in autonomous vehicles, which positively influenced their acceptance. Second, it suggests that the expected reduction in taxi drivers' workload due to autonomous vehic

---

## [Decision Letter · Decision Letter 1]

8 Oct 2024

PONE-D-24-30082R1Why Do People Resist AI-based Robotic Cars?: Analyzing the Impact of the Risk Perception Paradigm and Conditional Value on Public Acceptance of Autonomous Vehicles.PLOS ONE

Dear Dr. Kim,

Thank you for submitting your manuscript to PLOS ONE. After careful consideration, we feel that it has merit but does not fully meet PLOS ONE’s publication criteria as it currently stands. Therefore, we invite you to submit a revised version of the manuscript that addresses the points raised during the review process.

We look forward to receiving your revised manuscript.

Kind regards,

Zeashan Hameed Khan, Ph.D.

Academic Editor

PLOS ONE

Journal Requirements:

Reviewers' comments:

Reviewer's Responses to Questions

**Comments to the Author**

1. If the authors have adequately addressed your comments raised in a previous round of review and you feel that this manuscript is now acceptable for publication, you may indicate that here to bypass the “Comments to the Author” section, enter your conflict of interest statement in the “Confidential to Editor” section, and submit your "Accept" recommendation.

Reviewer #1: (No Response)

Reviewer #2: All comments have been addressed

2. Is the manuscript technically sound, and do the data support the conclusions?

Reviewer #1: (No Response)

Reviewer #2: Yes

3. Has the statistical analysis been performed appropriately and rigorously? 

Reviewer #1: (No Response)

Reviewer #2: Yes

4. Have the authors made all data underlying the findings in their manuscript fully available?

Reviewer #1: (No Response)

Reviewer #2: Yes

5. Is the manuscript presented in an intelligible fashion and written in standard English?

Reviewer #1: (No Response)

Reviewer #2: Yes

6. Review Comments to the Author

Reviewer #1: Well done, I recommend slight Minor English proofreading, it is necessary now.

Well done, I recommend slight Minor English proofreading, it is necessary now.

Reviewer #2: Thanks for providing detailed responses to my comments and making the necessary changes. The manuscript looks in good shape now and I would recommend acceptance.

7. PLOS authors have the option to publish the peer review history of their article (what does this mean?). If published, this will include your full peer review and any attached files.

Reviewer #1: **Yes: **Luttfi A. Al-Haddad

Reviewer #2: No

---

## [Author Response · Author response to Decision Letter 1]

14 Oct 2024

Comment from Reviewer 1 and Revision

□ Reviewer #1

□ Comment 1: Well done, I recommend slight Minor English proofreading, it is necessary now.

Well done, I recommend slight Minor English proofreading, it is necessary now.

○ Answer: We made the English proofreading. 

-We are very grateful for the comment reviewer gave. If reviewer gives us additional comments, we will revise it again. Thanks again for very good comment-

Comment from Reviewer 2 and Revision

□ Reviewer #2: 

□ Comment 1: Thanks for providing detailed responses to my comments and making the necessary changes. The manuscript looks in good shape now and I would recommend acceptance. 

○ Answer: Thank you for giving the good review for us. 

-We are very grateful for the comment reviewer gave. If reviewer gives us additional comments, we will revise it again. Thanks again for very good comment-

---

## [Editor Report · Decision Letter 2]

21 Oct 2024

Why Do People Resist AI-based Autonomous Cars?: Analyzing the Impact of the Risk Perception Paradigm and Conditional Value on Public Acceptance of  Autonomous Vehicles.

PONE-D-24-30082R2

Dear Dr. Kim,

We’re pleased to inform you that your manuscript has been judged scientifically suitable for publication and will be formally accepted for publication once it meets all outstanding technical requirements.

Kind regards,

Zeashan Hameed Khan, Ph.D.

Academic Editor

PLOS ONE

Additional Editor Comments (optional):

The paper has been improved and proof read after the first review cycle. Therefore, it can be accepted for publishing in the journal.
---

## [Editor Report · Acceptance letter]

13 Dec 2024

PONE-D-24-30082R2 

PLOS ONE

Dear Dr. Kim, 

I'm pleased to inform you that your manuscript has been deemed suitable for publication in PLOS ONE. Congratulations! Your manuscript is now being handed over to our production team.

Kind regards, 

on behalf of

Dr. Zeashan Hameed Khan 

Academic Editor

PLOS ONE